# Assessing Handrail-Use Behavior during Stair Ascent or Descent Using Ambient Sensing Technology

**DOI:** 10.3390/s23042236

**Published:** 2023-02-16

**Authors:** Yusuke Miyazaki, Kohei Shoda, Koji Kitamura, Yoshifumi Nishida

**Affiliations:** 1Department of Systems and Control Engineering, Tokyo Institute of Technology, 2-12-1, O-okayama, Meguro-ku, Tokyo 152-8550, Japan; 2National Institute of Advanced Industrial Science and Technology, 2-3-26, Aomi, Koto-ku, Tokyo 135-0064, Japan

**Keywords:** RGB-D camera, ambient sensing, frailty, stair ascending/descending

## Abstract

The increasing geriatric population across the world has necessitated the early detection of frailty through the analysis of daily-life behavioral patterns. This paper presents a system for ambient, automatic, and the continuous measurement and analysis of ascent and descent motions and long-term handrail-use behaviors of participants in their homes using an RGB-D camera. The system automatically stores information regarding the environment and three-dimensional skeletal coordinates of the participant only when they appear within the camera’s angle of view. Daily stair ascent and descent motions were measured in two houses: one house with two participants in their 20s and two in their 50s, and another with two participants in their 70s. The recorded behaviors were analyzed in terms of the stair ascent/descent speed, handrail grasping points, and frequency determined using the decision tree algorithm. The participants in their 70s exhibited a decreased stair ascent/descent speed compared to other participants; those in their 50s and 70s exhibited increased handrail usage area and frequency. The outcomes of the study indicate the system’s ability to accurately detect a decline in physical function through the continuous measurement of daily stair ascent and descent motions.

## 1. Introduction

Over the past few decades, the geriatric population has been increasing worldwide. The WHO states that between 2015 and 2050, the population of people with ages equal to and greater than 60 years will rise from 12% to 22% [1]. In Japan, the percentage of elderly was 28.8% as of 1 October 2020. By 2065, the age of one in every 2.6 people will be equal to or greater than 65, and that of one in 3.9 will be equal to or greater than 75 [2]. Elderly people rarely make a sudden transition from a healthy state to one that requires nursing care, often going through an intermediate stage referred to as frailty, which gradually leads to a state of dependency [3,4]. It has also been reported that a healthy state of a frail individual can be restored with appropriate interventions [5].

In Japan, a questionnaire-based frailty assessment, referred to as the “Kihon Checklist”, has been used as an indicator for frailty in the elderly [6,7]. This checklist consists of a group of 25 questions and is validated as a frailty index in Japan as well as in several other countries [8,9,10]. One question on this list is whether the person can ascend or descend stairs without using handrails or walls. Stair ascent or descent activities are known to be the most difficult physical task of daily activities for the elderly [11]. Therefore, assessing the characteristics of stair ascent and descent activities and handrail dependency in elderly people is significant to determine the frailty index.

However, the questionnaire-based frailty assessment requires the elderly to fill in the questionnaire voluntarily, regularly, and consciously, which is burdensome. Automated constant and spontaneous assessment of stair ascent and descent in daily life as well as quantitative frailty assessment based on motion measurement would help to reduce the burden and detect declines in physical functions in the elderly through long-term measurement.

To realize a system that can constantly and naturally assess stair ascent and descent in the daily life of elderly people, the Internet of Things (IoT) and machine learning (ML) have been employed. Markerless motion capture systems that include a depth camera are useful for the measurement of daily activities. Daily activities of the elderly have been captured using Microsoft Kinect v2 (Microsoft co.) to identify behavioral parameters that best distinguish between high- and low-fall risk individuals [12]. Kinect was deployed in the apartments of the elderly in an independent living facility to analyze gait characteristics by continuous in-home gait measurement [13]. Prediction of falls from pre-fall changes was explored based on the Kinect-recorded gait parameters over 10 years for the residents of independent living apartments [14]. A health monitoring system based on Kinect was developed to categorize movements during walking, standing up, and sitting down as normal or unusual [15]. Furthermore, a dataset of the daily activities of the elderly was developed, wherein the daily activities of 50 elderly were measured using Kinect v2 and classified into 55 actions [16]. We previously developed an “elderly behavior library” [17,18,19], which includes RGB-D videos, that showed the natural behavior of elderly people when using consumer products that were placed in their residences or residential facilities. We then analyzed the relationship between the natural standing behavior of elderly people and classes of standing aids as well as the physical and cognitive abilities that help prevent injury due to falling indoors [20,21]. Thus, the integration of ambient sensing into IoT devices has helped to record the daily activities of elderly people in their homes in a non-invasive manner and evaluate their physical functions according to the recorded patterns. However, to the best of the author’s knowledge, no studies have attempted to assess frailty based on the daily measurements of stair ascent and descent activities or dependence on the use of handrails. These parameters are closely related to frailty assessment.

Therefore, we developed an ambient sensing system for human behavior that could analyze the characteristics of ascending and descending stairs and handrail use behavior in daily life. The aim of this study was to construct an ambient sensing system that could non-invasively, continuously, and quantitatively assess stair ascending and descending characteristics for the elderly, and subsequently identify handrail gripping points. This study is the first to continuously measure the stair ascent and descent behavior as well as the handrail use in a natural living environment on a daily basis

## 2. Materials and Methods

### 2.1. Automatic Measurement of Three-Dimensional Human Behavior and Household Environment Using Depth Cameras

Azure Kinect DK (AK) (Microsoft Co.) was used as the depth camera. AK can estimate 32 three-dimensional (3D) skeletal points of the body skeleton in the camera coordinate system based on the depth information in conjunction with the Body Tracking Software Development Kit (SDK). In the case of long-term and continuous measurement of everyday activities with AK, it is ideal to store the skeletal coordinates only when a person appears in the angle of view. Therefore, we constructed a system that automatically stores the 3D coordinates of the skeleton and the RGB image of the depth camera viewpoint when a person appears and their skeleton is recognized. The sampling rate was 15 Hz.

### 2.2. Measurement Environment and Participants

#### Observation Environment for Stair Ascending or Descending Activities

The daily stair ascent or descent activities were measured in two separate houses, labeled Environment 1 and Environment 2. Environment 1 was a two-story house with four people living in it including a male in his 20s (p1), female in her 50s (p2), male in his 50s (p3), and a female in her 20s (p4). Environment 2 had two participants, a male in his 70s (p5) and a female in her 70s (p6). In Environment 1, a method for analyzing the stair ascent or descent activities (described in Section 2.3) was developed. The same method was integrated for analyzing participants in a pre-frail state in Environment 2.

In Environment 1, the AK recorder was installed on a staircase with a handrail, as shown in Figure 1. The three-dimensional motion was measured during staircase ascent and descent using the handrail (Figure 1a). The data acquired in this environment consisted of 402 ascending and descending activities. In Environment 2, the AK recorder was installed, as shown in Figure 1b. The data acquired in Environment 2 consisted of 22 ascent and descent activities. The participant labels (p1/p2/p3/p4/p5/p6), ascent and descent labels (up/down), and labels for objects held during ascent and descent (none/material/phone) were manually assigned to each series of obtained data. When analyzing the ascending and descending data, the data with the “none” label were included.

The study was conducted in accordance with the Declaration of Helsinki, and the protocol was approved by the Tokyo Institute of Technology Human Participants Research Ethics Review Committee (protocol code 2019150) and the AIST Ergonomics Experiment Review Committee (protocol code 2016-659H).

### 2.3. Defining Stair Ascent and Descent Data

#### 2.3.1. Definition of Stair Local Coordinate System

A local coordinate system that describes the stair ascent and descent motions were defined by setting the origin on the handrail and orthonormal–orthogonal basis from the depth map obtained from AK. Then, the stair ascending and descending motion data expressed in the camera coordinate system was transformed into a stair local coordinate system. This method helps compare human motions when ascending and descending stairs and behavioral patterns of using handrails in different environments.

#### 2.3.2. Obtaining 3D Coordinates

The 3D coordinates in the camera coordinate system of an arbitrary point on the depth map extracted from AK were obtained using the following approach: The depth map was obtained from AK in a 16-bit image format and then stored in each pixel. The 3D coordinates, p=[X,Y,Z], in real space for an arbitrary point C=[u,v] on the depth map obtained using the internal parameters ***A*** of the depth camera are used in Equation (1).
(1)$$\left(\begin{array}{c} u\\ v\\ 1 \end{array}\right)P_{z}=Ap$$
where $P_{z}$ represents the pixel value of the depth map. Subsequently, the three-dimensional coordinates in the environment were obtained using Equation (1).

#### 2.3.3. Orthonormal Basis for the Stair Coordinate System

The coordinate axes of the stair coordinate system were derived from the three-dimensional coordinate points of the handrails and walls of the staircase. The stair coordinate system shown in Figure 2 was defined based on the camera coordinate system. The longitudinal direction of the handrail in the right-hand coordinate system was defined as the *x*-axis of the stair coordinate system, the orthogonal downward direction as the z-axis, and the orthogonal lateral direction as the y-axis.

The start and end points of any line segment in the longitudinal direction of the handrail were selected from the depth map and converted to three-dimensional coordinates using Equation (1) to obtain the longitudinal vector $n_{x}$ of the handrail.

Next, the *y*-plane and its normal vector ${n_{y}}^{\prime }$ were obtained using the following approach. A rectangular region was manually selected from the depth map wall surface, and the 3D coordinate point cloud P=[X,Y,Z] in the region was obtained using the same approach. ***P*** follows Equation (2).
(2)$$\left[X,Y,1\right]\left[\begin{array}{c} a\\ b\\ d \end{array}\right]=-cZ+E$$
where ***E*** is the distance from the *y*-plane to each point. In this case, because the plane that is measured by the camera is not always perpendicular to the camera, constants *a* and *c* and constants *a*, *b*, and *d* are assumed so that *c* ≠ 0.

Solving this using the least-squares method yields Equation (3), where $P^{\prime }=\left[X,Y,1\right]$, so we obtain the following.
(3)$$\left[\begin{array}{c} a\\ b\\ d \end{array}\right]=-{\left({P^{\prime }}^{T}P^{\prime }\right)}^{-1}{P^{\prime }}^{T}Z$$

The wall normal vector ${n_{y}}^{\prime }$ can be defined using *a* and *b*, as shown in Equation (4).
(4)$${n_{y}}^{\prime }=\left[\begin{array}{c} a\\ b\\ c \end{array}\right]$$

Then, the vectors $n_{x}$ and ${n_{y}}^{\prime }$ are normalized, and $n_{z}$ is calculated from Equation (5).
(5)$$n_{z}=n_{x}\times {n_{y}}^{\prime }$$

Because ${n_{y}}^{\prime }$ is not always orthogonal to $n_{z}$ and $n_{x}$, the value of $n_{y}$ can be determined again using Equation (6) to define an orthonormal basis for the staircase coordinate system.
(6)$$n_{y}=n_{z}\times n_{x}$$

The edge point of the handrail on the depth map was selected as the stair coordinate system origin $P_{0}$. A homogeneous coordinate transformation matrix was then defined using $P_{0}$ and the orthonormal basis of the staircase coordinate system. The rotation matrix R is defined using Equation (7) according to $n_{x},n_{y},n_{z}$, and the homogeneous transformation matrix H is defined using Equation (8). The stair ascent and descent motions measured in the camera coordinate system were transformed into the stair coordinate system using ***H***.
(7)$$R=\left[\begin{array}{c} n_{x}{}^{T}\\ n_{y}{}^{T}\\ n_{z}{}^{T} \end{array}\right]$$
(8)$$H=\left[\begin{array}{cc} R & P_{0}\\ 0 & 1 \end{array}\right]\;$$

#### 2.3.4. Calculating Stair Ascent and Descent Speed

Kawai et al. demonstrated that walking speed in daily life reflects physical function and that walking speed may be used to screen frailty [22]. Therefore, measuring the speed of ascending and descending stairs in daily life helps assess frailty. Each participant’s stair ascent/descent speed was defined as the average speed in the range *x* = 0–1000 mm using the pelvis point. T-tests were conducted to estimate the difference between the ascent and descent speeds within each participant.

### 2.4. Determining Handrail Grasping Point

To quantitatively assess the dependence on handrails, which is one of the basic checklist items for frailty assessment, we developed a method to calculate the handrail grasping ratio when ascending and descending stairs. The Azure Kinect Body Tracking SDK defines three skeletal coordinates for each hand: HAND, HANDTIP, and THUMB [23]. The respective coordinates were $P_{hand},\;$
$P_{handtip}$ and $P_{thumb}$, and it was assumed that the grasping point coincides with the center-of-gravity coordinates$\;P_{gp}$ of those points.
(9)$$P_{gp}=\frac{1}{3}\left(P_{hand}+P_{handtip}+P_{thumb}\right)=\left(\begin{array}{c} \begin{array}{c} P_{gpx}\\ P_{gpy} \end{array}\\ P_{gpz} \end{array}\right)$$

Among the 402 ascent and descent motions, 10 were extracted as sample motions, one with and another without grasping the handrail. The images stored in the time frames of each motion were manually checked and labeled as either “grasped” or “not grasped” (Figure 3).

After extracting the motions in the range 0 < $P_{gpx\;}$< 1000 as the handrail’s straight part, a decision tree model was constructed for the classification of labels as either “grasped” or “not grasped” by using $P_{gpy}\;$and $P_{gpz}$ as the predictors. Table 1 shows the resulting confusion matrix that was trained using the 5-segment cross-validation. The model yielded a classification accuracy of 86.9% (Figure 4). A grasp classification model for Environment 2 was also created for the ascending and descending data in p5–p6 using the same method, as shown in Figure 5. The decision tree model with a classification accuracy of 85.2% was used as the grasp classification model (Table 2).

### 2.5. Evaluating Handrail Grasping Area and Frequency

The ratio of handrail grasping in the stair ascent and descent activities was calculated to evaluate the degree of dependence on handrails in the assessment of frailty. The grasp classification model was applied to all ascending and descending activities without a single HAVE_LABEL, and the ratio of frames to be grasped for the total number of frames (n) was calculated for each participant.

In addition, the area and frequency of the handrail grasping were visualized as a heat map. The developed method highlighted the grasping points of the handrail by transforming $\;P_{gp}$ to the camera coordinate system in the frames where grasping was estimated and then projected it onto the RGB image using the perspective projection method.

## 3. Results

### 3.1. Stair Ascent and Descent Speed

As shown in Figure 6 and Table 3, the average ascent speeds of p1–p4 were within the range of ±10% and were almost identical. Conversely, the ascent speeds of p5 and p6 were less than the ascent speed of p1 at a significance level of 1%. As shown in Figure 7, the descent speeds of p1–p4 were almost the same, and those of p5 and p6 were lower than that of p1 at a significance level of 5%.

In addition, the speeds of ascent and descent tended to differ according to the participants, with descent speeds higher than the ascent speeds in the cases of p1–p5, while the descent speed of p6 was significantly slower than the ascent speed at a significance level of 5%.

### 3.2. Handrail Grasping Ratio and Grasping Area

From the handrail grasping ratios shown in Table 4 and Table 5 and the handrail grasping area visualized in Figure 8, p1, a male in his 20s, showed the smallest handrail grasping ratio when ascending and descending stairs in Environment 1. Therefore, p1 almost entirely ascended and descended the stairs without using a handrail, indicating that he is not dependent on a handrail. Second, p3, a male in his 50s, was less dependent on the handrail, and his handrail grasping ratio tended to increase slightly when descending. P4, who is in her 20s, held the handrail more while ascending than while descending. However, her handrail grasping position was concentrated locally, which indicates her preference for using the handrail. Furthermore, p2, who is in her 50s, held the handrail the most among the participants in Environment 1, both when ascending and descending. The grasping ratio increased approximately twice as much when descending, and the grasping positions were evenly distributed throughout the handrail, indicating an increase in handrail dependence.

The handrail grasping ratios of p5–p6 in their 70s in Environment 2 were generally higher than those of the participants in their 20s (p1, p4) and 50s (p2, p3) in Environment 1. Furthermore, p5 and p6 grasped the handrail for approximately 80% of the time when ascending, and almost all areas of the handrail were red, indicating that they were significantly dependent on the handrail when ascending the stairs. When descending the stairs, the handrail dependence of p5 decreased slightly, while that of p6 increased further, indicating that p6 tended to depend on the handrail even more.

## 4. Discussion

We developed a method to quantitatively assess the stair ascent/descent speed and handrail dependence during stair ascent/descent, with the aim of frailty assessment [6,9,10,24]. Conventional frailty assessments were conducted by a self-completion questionnaire that was performed based on measurements of daily activities in homes. Although the physical function during stair ascent and descent has been conventionally evaluated in a laboratory environment and a detailed biomechanical evaluation is possible [25,26,27,28], to the best of the author’s knowledge, this study is the first to continuously measure the stair ascent and descent behavior as well as the handrail use in a natural living environment on a daily basis. In some studies [29,30], an inertial measurement unit was attached to the body to serve as an approach that integrated wearable sensors. However, the systems cannot assess the degree of handrail dependence, which is important for frailty assessment. The developed system used both the 3D human skeletal information and point cloud information of the environment based on the information recorded using an RGB-D camera. This further ensures that the system can measure human movements and analyze the relationship between the environment and product use behavior (i.e., the degree of handrail dependence).

The measured ascent and descent speeds of p5–p6 in their 70s were significantly less than those of p1–p4. In addition, p5 and p6 can be presumed to be in a pre-frail state, given that each of their answers to the “Kihon Checklist” had a score of four. This indicates that the physical functions of p5 and p6 were less compared to those of the participants in their 20s and 50s. Furthermore, the handrail grasping ratios of p5 and p6 tended to be higher than those of p1–p4.

In addition, the descent speed of p6 was significantly less than the ascent speed. Reeves et al. showed that elderly people may meet the demands of unaided stair ascent by adopting several alternative strategies to compensate for their reduced musculoskeletal capabilities [31]. The stair descent activity helps improve physical functions because of the high muscular load [25,28]. Therefore, it is important to focus on the descent movement and evaluate the degree of dependence on the handrails for frailty assessment. In Environment 1, the grasping ratio was higher in p2, a female participant in her 50s, than in the other participants. Furthermore, her handrail dependence was particularly high when descending. Therefore, a quantitative evaluation of the handrail dependence of the physical functions of participants who were not in a state of pre-frailty and were considered to be healthy at the present stage can represent the potential level of frailty. The results indicate that it is important to distinguish between the ascent and descent movements and that the degree of dependence on handrails during the descent can be used to evaluate frailty.

### Limitations and Future Scope

This study developed a system that can assess the daily stair ascent and descent activities and the degree of dependence on handrails. Long-term measurement is necessary to detect a decline in physical function from the daily stair ascent and descent activities. In this study, measurements were taken over a period of two months. Nevertheless, even longer periods of measurement will be required to detect a decline in physical function. To capture changes in physical function over a long period of time, it is necessary to develop a detection system based on machine learning that can detect changes in handrail dependence and automatically diagnose frailty in the daily environment.

Additionally, the relationship between stair and handrail design and stair ascent and descent characteristics should be clarified by conducting measurements in a variety of environments in the future. This will enable a suitable product design of stairs and handrails to accommodate changes in the physical characteristics of the elderly.

Furthermore, the system could be expanded to biomechanical assessment. In this study, we focused on assessing the stair ascending/descending speed and handrail dependence corresponding to frailty assessment using the basic checklist, a self-administered questionnaire currently in use. However, because the system acquires the skeletal coordinates of the entire body, it will be possible to analyze the feature motions of stair ascent and descent motions in the future. The stair ascent and descent activities are known to be the most difficult physical tasks in daily activities [11], and thus they have been studied widely under laboratory conditions [27,32,33,34,35]. However, it is difficult to reproduce the natural behavior of the elderly in a controlled laboratory environment. The system developed in this study will enable a detailed assessment of frailty based on the biomechanical analysis of actual stair ascent and descent activities in a real-life environment.

## 5. Conclusions

This study developed a system to automatically and continuously measure and analyze the ascent and descent motions and handrail-use behaviors in an actual ambient living environment. An RGB-D camera was used in addition to the conventional questionnaire-based frailty assessments. Daily stair ascent and descent motions were measured in two separate houses, with two participants in their 20s and two in their 50s in the first house, and two participants in their 70s in the second house. The outcomes of the study indicated that the participants in their 70s exhibited a decreased stair ascent/descent speed compared to the other participants, and those in their 50s and 70s exhibited an increased handrail usage area and frequency, particularly during descent. The results indicate the potential of this system to detect physical frailty by ambient and continuous measurement of the daily stair ascent and descent motions.

## Figures and Tables

**Figure 1 sensors-23-02236-f001:**
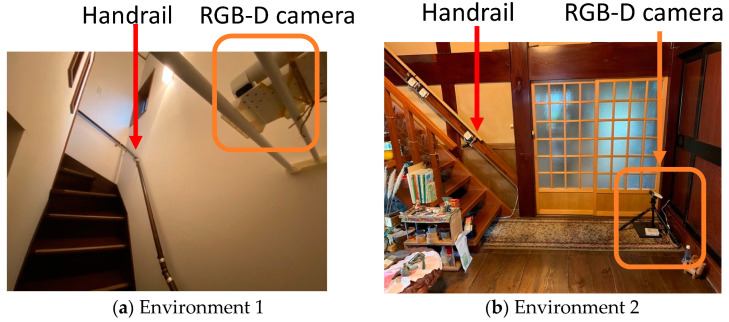
Stair, handrail, and RGB-D camera in the measurement environment.

**Figure 2 sensors-23-02236-f002:**
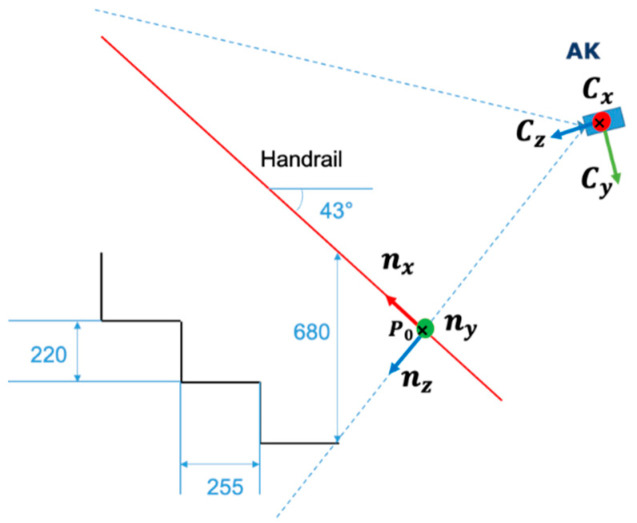
Stair coordinate system for Environment 1.

**Figure 3 sensors-23-02236-f003:**
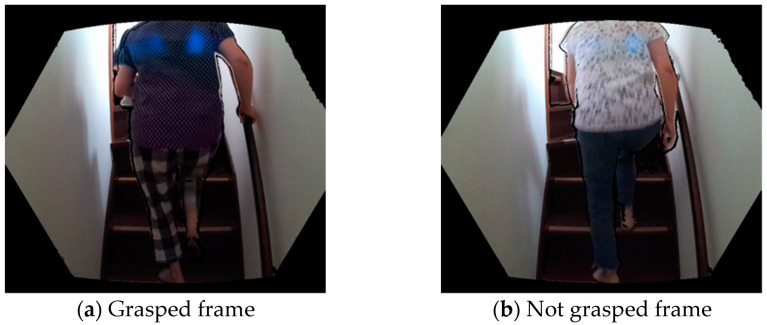
Labeling of the “grasp” or “not grasped” frame.

**Figure 4 sensors-23-02236-f004:**
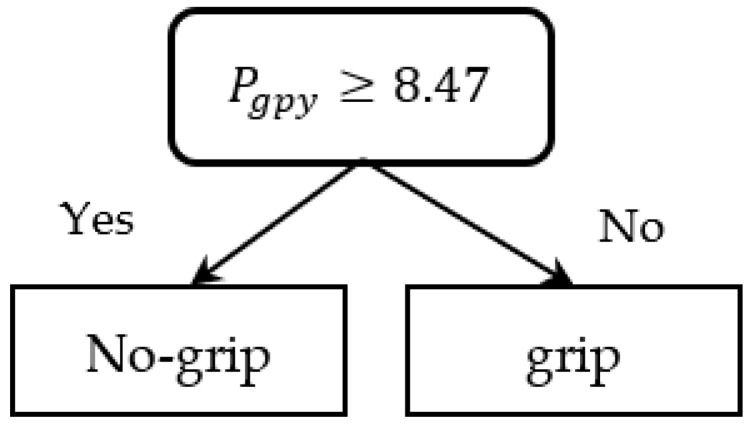
Grasping classification decision tree model in Environment 1.

**Figure 5 sensors-23-02236-f005:**
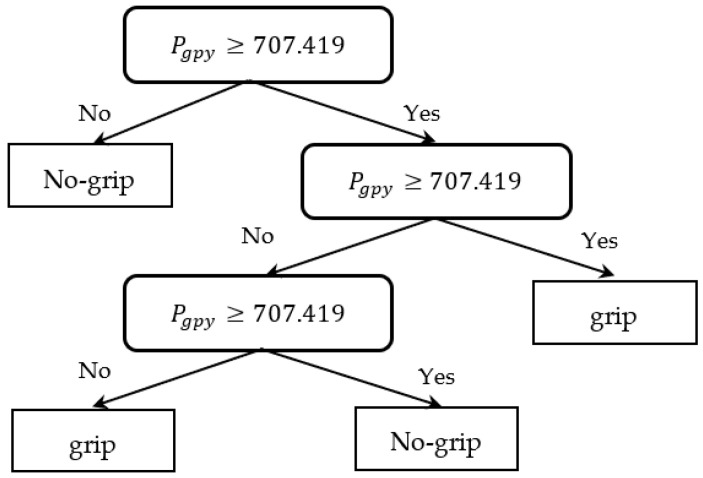
Grasping classification decision tree model in Environment 2.

**Figure 6 sensors-23-02236-f006:**
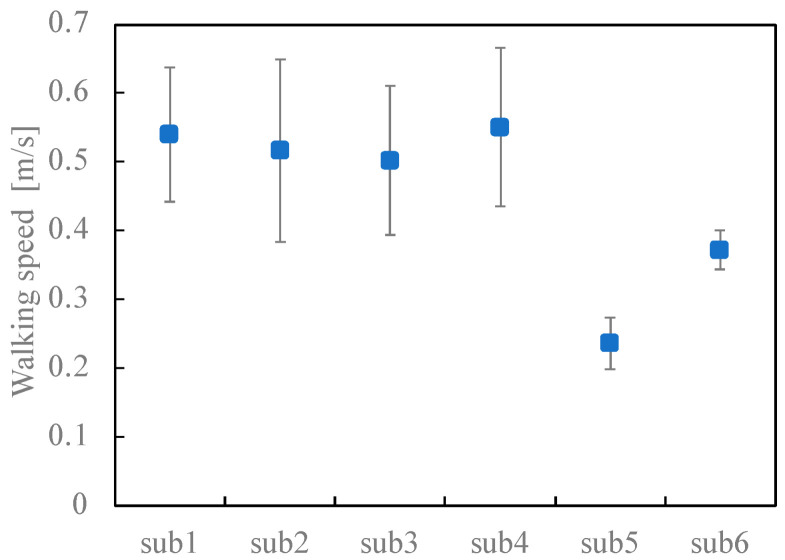
Comparison of the ascent speed between participants. Blue circles and error bars indicate the mean values and standard deviations, respectively.

**Figure 7 sensors-23-02236-f007:**
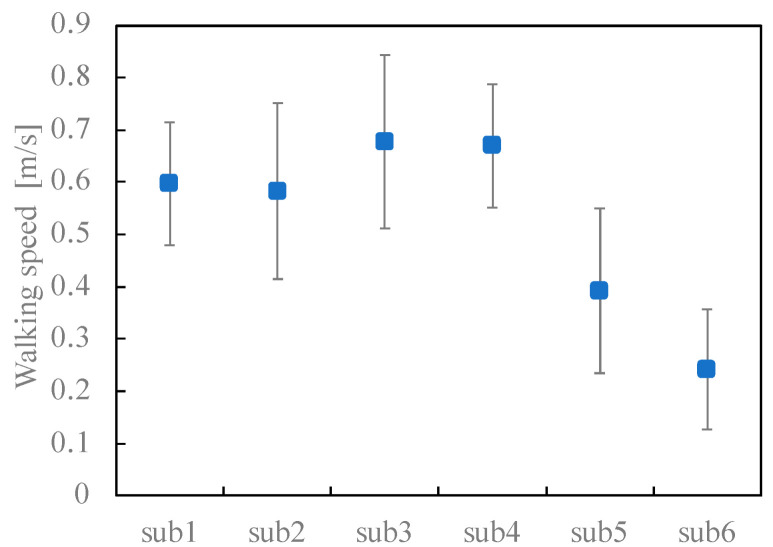
Comparison of the descent speed between participants. Blue circles and error bars indicate the mean values and standard deviations. respectively.

**Figure 8 sensors-23-02236-f008:**
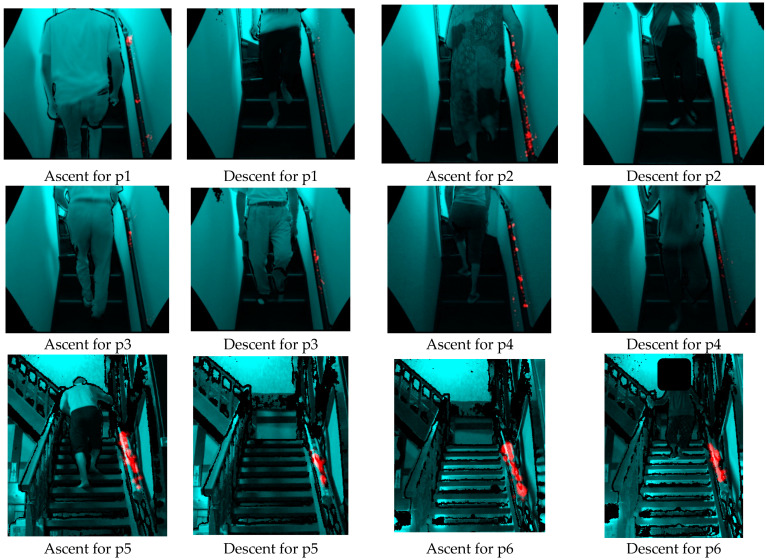
Heat map of the grasping points. Red color indicates a more intense area where the participant grasped more frequently.

**Table 1 sensors-23-02236-t001:** Confusion matrix for the decision tree model in the case of Environment 1.

	Predicted	
	Non Grip	Grip
Non grip (*n* = 139)	117	22
Grip (*n* = 136)	14	122

**Table 2 sensors-23-02236-t002:** Confusion matrix for the decision tree model in the case of Environment 2.

	Predicted	
	Non-Grip	Grip
Non-grip (*n* = 41)	29	12
Grip (*n* = 74)	5	69

**Table 3 sensors-23-02236-t003:** Stair ascent and descent speed for each participant.

Participant	Ascent (m/s)			Descent (m/s)			*p*
	*n*	Mean	SD	*n*	Mean	SD	
p1	84	0.5398	0.0976	88	0.5970	0.1185	<0.01
p2	38	0.5164	0.1326	38	0.5828	0.1676	0.0593
p3	57	0.5024	0.1089	52	0.6763	0.1664	<0.01
p4	20	0.5502	0.1151	20	0.6698	0.1177	<0.01
p5	5	0.2371	0.0376	7	0.3916	0.1573	0.0595
p6	5	0.3722	0.0285	5	0.2419	0.1137	<0.05

**Table 4 sensors-23-02236-t004:** Grabbing ratio in the ascent motion.

p1 (*n* = 2126)	p2 (*n* = 856)	p3 (*n* = 1406)	p4 (*n* = 633)	p5 (*n* = 103)	p6 (*n* = 135)
4.6%	25%	2.5%	19%	81.6%	76.3%

**Table 5 sensors-23-02236-t005:** Grabbing ratio in the descent motion.

p1 (*n* = 1193)	p2 (*n* = 382)	p3 (*n* = 565)	p4 (*n* = 365)	p5 (*n* = 85)	p6 (*n* = 84)
3.8%	51%	8.7%	9.3%	48.2%	83.3%

## Data Availability

Not applicable.

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
