# Peer review of "Assessing Handrail-Use Behavior during Stair Ascent or Descent Using Ambient Sensing Technology"

_sensors, 2023, doi:10.3390/s23042236_

Round 1
Reviewer 1 Report
the work is dedicated to an interesting topic and the motivation for conducting the study is also comprehensible - instead of the (typically falsified) answering of a questionnaire, a relevant condition is automatically collected - for example the use of stairs.
The work is basically okay and of interest to readers, but there are some weaknesses in the details. The wording needs to be revised, there are some typos, inconsistencies in capitalization and double words.
In terms of content, the motivation part is not entirely coherent. A collection of ML work is cited, from very broad fields of applications. Before these contents, a subheader (Related Work, Focus..) would make sense. I would put the following sentence at the beginning: "The following studies demonstrate the significance of the Internet of Things (IoT) and ML in assessing and recording the daily activities of the elderly". and then enumerate the studies, try to structure/classify the studies into fields of applications and strengthen the motivation why the combination of IoT / ML makes sense in the context of the presented work. In my opinion, it would be important to limit the focus of work. In the related work dementia is thematized, but I see it quite difficult to cover such long-term aspects in a short-term evaluation.
In regard to the results it is not really surprising that the elderly persons in the sample show a higher degree of frailty, use the handrail more often and need a longer time to pass the stairs. In the discussion, I therefore miss - besides the mentioned increase of the sample size in future - the aspect of long-term evaluation such that changes in the observed behaviour (usage of stairs) can be analyzed and the development of physical as well as mental impairments could be recognized.
Reviewer 2 Report
The paper is clearly written and well-documented. The organization is good. The presentation is good. References are good. The readability is excellent. Although more detailed performance considerations would improve the paper's value, it seems to me suitable to be accepted for publication in this journal.
There are a few issues that, as mentioned in the comments below, would benefit from clarification or further discussion before the paper is accepted:
1) The introduction offers a good, broad entrance into the topic. It involves many sources of scientific literature. However, a background section prior to Section 2, "Materials and Methods," is required to inform readers of the related works that are directly related to ambient sensing systems for human behavior as well as the differences between other systems. It would also help to expand the references.
2) It is necessary to improve the quality of figures 2, 5, and 6, as their legibility is almost non-existent. In addition, the format of tables 1 and 2 does not comply with what is stipulated by the journal.
3) In addition, there are a few problems with the format of the manuscript, so the author should carefully modify it according to the standards of the journal.
Round 2
Reviewer 1 Report
The authors have given due consideration to my comments and have revised the paper. Therefore, I agree to publication.
Reviewer 2 Report
I agree with the changes made. However, I notice some double spacing throughout the text. If they are corrected, it should be possible to avoid having certain figures on one page and their descriptions on the next page (i.e., Table 2—Line 199, Figure 3—Line 84, and Figure 1—Line 114).